# How Does the Pandemic Facilitate Mobile Payment? An Investigation on Users’ Perspective under the COVID-19 Pandemic

**DOI:** 10.3390/ijerph18031016

**Published:** 2021-01-24

**Authors:** Yuyang Zhao, Fernando Bacao

**Affiliations:** NOVA Information Management School (NOVA IMS), Campus de Campolide, Universidade Nova de Lisboa, 1070-312 Lisboa, Portugal; bacao@novaims.unl.pt

**Keywords:** unified theory of acceptance and use of technology (UTAUT), mental accounting theory (MAT), mobile payment, adoption intention, COVID-2019

## Abstract

Owing to the convenience, reliability and contact-free feature of Mobile payment (M-payment), it has been diffusely adopted in China during the COVID-19 pandemic to reduce the direct and indirect contacts in transactions, allowing social distancing to be maintained and facilitating stabilization of the social economy. This paper aims to comprehensively investigate the technological and mental factors affecting users’ adoption intentions of M-payment under the COVID-19 pandemic, to expand the domain of technology adoption under the emergency situation. This study integrated Unified Theory of Acceptance and Use of Technology (UTAUT) with perceived benefits from Mental Accounting Theory (MAT), and two additional variables (perceived security and trust) to investigate 739 smartphone users’ adoption intentions of M-payment during the COVID-19 pandemic in China. The empirical results showed that users’ technological and mental perceptions conjointly influence their adoption intentions of M-payment during the COVID-19 pandemic, wherein perceived benefits are significantly determined by social influence and trust, corresponding with the situation of pandemic. This study initially integrated UTAUT with MAT to develop the theoretical framework for investigating users’ adoption intentions. Meanwhile, this study originally investigated the antecedents of M-payment adoption under the pandemic situation and indicated that users’ perceptions will be positively influenced when technology’s specific characteristics can benefit a particular situation.

## 1. Introduction

With the increasingly widespread popularity of mobile devices, our daily lives have significantly changed, especially in terms of financial transactions. Mobile payment (M-payment) has been dramatically adopted in various industries in recent years. According to a WorldPay report, M-payments accounted for 22% of the global points of sale spending in 2019, and this percentage will increase to 29.6% in 2023 [1]. Moreover, China’s overwhelming adoption of M-payments (Alipay and Wechat Pay) at the point of sale by using Quick Response (QR) codes drove nearly half (48%) of the point-of-sale payments in 2019 [1]. Various previous studies have facilitated the understanding of adoption intentions of M-payment in different contexts [2,3,4]. However, there are still deficiencies of determinant variation and theoretical evidence of different perspectives in emergency conditions [5].

The 2019 novel coronavirus (COVID-2019) broke out in December of 2019 and has dramatically expanded globally. As of 7 December 2020, there were 66,243,918 confirmed cases of COVID-19 and 1,528,984 deaths worldwide, reported by the World Health Organization [6]. Due to the high risk of COVID-19 transmission, reducing contact among people and maintaining social distancing was highly recommended by the WHO (2020b) and Tang et al. (2020) [7,8]. In this sense, the contactless characteristic of M-payments can potentially contribute to users’ mental and physical expectations to support their transaction processes and protect their safety. Accordingly, adoption of M-payment in China has significantly increased during the COVID-19 pandemic. According to a report from China banking and insurance news (2020), during the COVID-19 pandemic, the number of transactions made by M-payment was 22.4 million in the first quarter of 2020 in China, up 187% from the previous year (2019) [9]. Meanwhile, based on the CNNIC (2020) report comparing the smartphone users who used M-payment from 2019 to 2020, this percentage increased from 73.5% in June 2019 to 85.3% in March 2020 and reached 86.0% in June 2020 in China, which indicates that M-payment contributes to maintaining individual and organizational transactions during emergency situations [10]. Furthermore, users’ payment habits and business models have changed from traditional face-to-face transactions to contactless M-payment transactions during the pandemic, which in turn efficiently supports the survival of various business and maintains the development of the social economy under an emergency situation. Therefore, what factors influence users’ intentions to adopt M-payment during the pandemic? It becomes dramatically valuable to understand customers’ behaviors under the pandemic for relevant researchers and stakeholders to comprehensively investigate information technology adoption under an emergency situation to develop business strategies correspondingly.

Traditional adoption models (e.g., Technology Acceptance Model (TAM) and the Unified Theories of Acceptance and Use of Technology (UTAUT)) evaluate users’ intentions determined by technological perceptions with an obvious limitation of influence from users’ mental perceptions [11,12]. Notably, based on the recommendations of governments and the WHO (2020b) regarding restrictions of direct and indirect contacts among people under the pandemic situation [7], the contactless feature of M-payment potentially influenced users’ attitudes regarding the benefits of using M-payment for daily transaction, which indicates that environmental conditions affect users’ mental process with regard to adopting M-payment [13]. Thus, this paper involved mental accounting theory (MAT) to explain customers’ psychological cognitions of the benefits of using M-payment under a pandemic situation. In order to fill the gap of limited integration of technological and mental perceptions on technology adoption, this study incorporates MAT with UTAUT to comprehensively investigate the antecedents of M-payment adoption on users’ perspectives. Specifically, perceived benefits are considered as an important factor in terms of users’ expectations and will help determine their decisions [14]. Meanwhile, due to the influence of the pandemic, perceived security and trust are also considered as additional antecedents of users’ adoption intentions of M-payment [15]. Perceived security is the most significant determinant of trust, positively affecting users’ intentions of using M-payment [16]. Therefore, this study proposes a new adoption model, including perceived benefits, performance expectancy, effort expectancy, social influence, perceived security, trust and behavioral intention, to investigate users adopting M-payment during the COVID-19 pandemic in the following sections: Section 2: theoretical backgrounds of the utilization of M-payments during the COVID-19 pandemic, MAT and UTAUT; Section 3: development of hypotheses and research model; Section 4: research methodology and data demonstration; Section 5: data analysis; Section 6: discussion; Section 7: theoretical and practical implications; Section 8: limitations and future research recommendations; Section 9: conclusions.

## 2. Theoretical Background

### 2.1. M-Payment and Its Utilization under the COVID-19 Pandemic

M-payment, as an information interaction electronic financial transaction method for paying goods, services and bills by mobile devices [5], consists of three leading contactless technologies, including Short Message Service (SMS), Near Field Communication (NFC) and Quick Response (QR) codes [2]. Due to the convenient, open and secure features of M-payment, a new business climate has been formulated by the wide adoption of M-payment, as financial transactions, are able to take place anywhere, anytime and by anyone, which has established colossal market potential in various contexts, especially under pandemic situations [17]. Many researchers have investigated various factors affecting M-payment adoption by reviewing theoretical frameworks and variables, supporting that relevant knowledge and understanding of users’ adoption intentions of M-payment is determined by technological and mental perceptions, as shown in Table 1. However, few studies have analyzed the adoption intentions determined by mental and technological factors conjointly under an emergency situation.

COVID-19, as a global pandemic, has dramatically influenced people’s daily lives and the world economy. According to relevant studies [8] and a report from the WHO (2020b), COVID-19 has significant transmission risk by direct contact with infected people and indirect contact with surfaces in the immediate environment or with objects used on an infected person. In this sense, the contact rate can significantly contribute to the infection risk of COVID; thus, the contactless feature, as a typical characteristic of M-payment, provides mental and physical support to protect and maintain users’ experience in transactions [20]. Moreover, due to the restrictions imposed by the Chinese Government to avoid direct contact and maintain social distancing during the COVID-19 pandemic, M-payment had been widely adopted for its contactless feature and trustworthy performance. Users’ positive cognitions and feelings of safety when using M-payment as the main payment method have been formulated, which reduces the virus transmission risk, protects personal safety and supports the social economy [9].

### 2.2. Mental Accounting Theory (MAT)

Mental accounting theory (MAT), proposed by Thaler (1985), is defined as the set of individuals’ cognitive operations to categorize, organize and evaluate the consequences of their decision-making in financial activities [21]. Specifically, MAT explains that personal desires influence the cognitive processes of individuals, and their psychological processes for valuing a specific technology should be taken into consideration in the environment of voluntary usage [22]. Accordingly, based on the normative principle of fungibility at the point of purchase, mental accounting is engaged, and decision-making is based on the evaluation of perceived benefits of the purchase activity [23]. Concretely, in the technology adoption aspect, a consumer’s decision of adoption is based on the perceived benefits of utilization of technology [13]. Moreover, MAT can also be incorporated into an adoption model to complementarily explain customers’ intentions of technology adoption [24]. Cheng and Huang (2013) incorporated MAT into TAM to investigate the mental factors affecting customers’ intentions of adopting high-speed railway mobile ticketing services [25]. Park et al. (2018) proposed that the multidimensional perceived benefits of M-payment services are influenced by social influence and technology anxiety, which indicates that users’ willingness of using M-payment is significant determined by the external environment and internal technological perception [14]. Furthermore, MAT provides a theoretical basis to explain consumers’ decisions under conditions of risk and uncertainty [13]. Combined with the disaster of COVID-19, customers’ psychological processes of adopting M-payment are significantly influenced by the contactless feature of M-payment, which is appropriately adapted to the environmental situation, public restriction and users’ requirements. Therefore, MAT is appropriate to apply for explaining users’ mental cognitions of using M-payment under the COVID-19 pandemic.

### 2.3. Unified Theory of Acceptance and Use of Technology (UTAUT)

UTAUT was developed by Venkatesh et al. (2003). It consists of performance expectancy, effort expectancy, social influence and facilitating conditions as determinants of behavioral intentions to use a new technology system [26]. UTAUT has been applied in various contexts of technology adoption. It has been revised with additional variables to explain users’ behavioral intentions [4]. For example, Khalilzadeh et al. (2017) integrated security-related factors with the UTAUT model and validated that security and trust have a strong effect on customers’ adoption intentions of NFC M-payments in the restaurant industry [15]. Marinković et al. (2020) modified the UTAUT model with extra variables (perceived trust and satisfaction) to evaluate customers’ usage intentions of M-commerce [27]. Moreover, UTAUT has also been integrated with other models to evaluate users’ behavioral intentions [2,28]. Di Pietro et al. (2015) integrated TAM, DOI and UTAUT to verify M-payment adoption intentions [2]. Oliveira et al. (2014) integrated UTAUT with the initial trust model and task–technology fit model to investigate users’ behavioral intentions of adopting mobile banking in Portugal [28]. However, UTAUT focuses on technological expectations rather than mental expectations, which weakly explains users’ expectations determining their intention of technology [12]. Thus, it is necessary to integrate UTAUT with MAT to explain users’ technological and mental perceptions complementarily on usage intention of M-payment during the COVID-19 pandemic. The development of hypotheses and research models is illustrated in the following section.

## 3. Development of Hypotheses and Research Model

### 3.1. Revisiting the MAT

#### Perceived Benefits (PBs)

According to MAT, when consumers perform a particular behavior, they tend to evaluate a possible beneficial outcome [21]. Perceived benefits represent users’ perceptions of the functional benefits of M-payment services, which determine their decisions of adoption [14]. Perceived benefits support a better understanding of users’ mental perceptions of adoption intentions in various technologies, such as online shopping [29], and mobile banking [30]. Meanwhile, perceived benefits have been identified as multidimensional benefits, including utilitarian, hedonic and social values, which are determined by social influence and technology uncertainty [14,24]. However, few studies focus on the perceived benefits of technology characteristics corresponding to a particular condition. Specifically, in a pandemic situation, social distancing is an efficient way to decrease COVID-19 transmission risk among people [7,31]. Compared with traditional payments, the contactless characteristic of M-payments supports users in maintaining social distancing to avoid direct and indirect contacts from cash or point of sale terminals during a transaction process. This aspect allows users to formulate their opinions on the perceived mental and physical benefits of personal safety and provides convenience and utility when using M-payment technology as a financial transaction method in the COVID-2019 pandemic. Thus, perceived benefits are considered as a mental factor to influence the users’ adoption intentions of M-payment during the COVID-19 pandemic, expressed as the following hypothesis.

**Hypothesis** **1.**
*Perceived benefits have a positive effect on the behavioral intentions to adopt M-payments during the COVID-19 pandemic.*


### 3.2. Revisiting UTAUT

#### 3.2.1. Performance Expectancy (PE)

Performance expectancy is defined as an individual’s perception in terms of the use of an information system facilitating the completion of a task and work performance [26]. Performance has been conceptualized by using attributes related to the system’s efficiency, speed and accuracy in task completion [11]. Especially during the COVID-19 pandemic, users show more concern toward payment efficiency and accuracy. Concretely, in the M-payment adoption aspect, performance expectancy has significantly positive effects on users’ adoption intentions in various contexts [2,3,32,33]. Therefore, when users perceive M-payment as a useful way to accomplish their transactions during the pandemic, they will choose M-payment instead of traditional payment. Accordingly, this paper proposes the following hypothesis.

**Hypothesis** **2.**
*Performance expectancy has a positive effect on the behavioral intention to adopt M-payments during the COVID-19 pandemic.*


#### 3.2.2. Effort Expectancy (EE)

According to UTAUT, effort expectancy is referred to as “the degree of ease associated with the use of the system” [26]. Effort expectancy influences users’ attitudes toward adopting M-payment [17], revealing an even higher influence than performance expectancy [34]. Specifically, Liébana-Cabanillas et al. (2018) found that effort expectancy is the most significant factor affecting users’ intentions of using NFC M-payment systems in public transportation [3]. Moreover, effort expectancy has also been verified to have a positive impact on performance expectancy in various technology adoption contexts [2,17,35]. Therefore, the following hypotheses are proposed.

**Hypothesis** **3.**
*Effort expectancy has a positive effect on the behavioral intention to adopt M-payments during the COVID-19 pandemic.*


**Hypothesis** **4.**
*Effort expectancy has a positive effect on the performance expectancy to adopt M-payments during the COVID-19 pandemic.*


#### 3.2.3. Social Influence (SI)

In terms of UTAUT, the definition of social influence is “the degree to which an individual perceives that significant others believe he or she should use the new system” [26]. Slade et al. (2015) explained that it is an underlying assumption that users prefer to consult their social network to reduce any anxiety arising from uncertainty [36]. Especially during the COVID-19 pandemic, recommendations and suggestions from important, relevant people are more important for individuals’ decisions and actions. From previous studies, social influence has been widely tested in the different contexts of its impact on usage intention of mobile technologies [15,24,33,36]. Morosan and DeFranco (2016) presented that social influence has a significant effect on the intention of using M-payment [37]; Kerviler et al. (2016) illustrated that social influence plays a considerable role in explaining users’ intentions of using M-payment [24]. Moreover, social influence, as a determinant for formulating users’ attitudes, significantly affects the perceived multibenefits of users with regard to using M-payment services [14]. Thus, relevant hypotheses are proposed as follows.

**Hypothesis** **5.**
*Social influence has a positive effect on the behavioral intention to adopt M-payments during the COVID-19 pandemic.*


**Hypothesis** **6.**
*Social influence has a positive effect on the perceived benefits to adopt M-payments during the COVID-19 pandemic.*


#### 3.2.4. Trust (TR)

Trust is defined as users’ willingness to expect a positive outcome of technology’s future performance and a subjective belief that the service provider will fulfil their obligations [38]. Meanwhile, the COVID-19 pandemic has brought uncertainty and social pressure to individuals’ daily transaction processes. Trust of M-payment platforms can increase the likelihood of users using them to make contactless M-payments rather than traditional payments [27,39]. Zhu et al. (2017) validated that trust has the most significant effect on the behavioral intention to use M-payment [39]. Meanwhile, many studies have also verified the effect of trust significantly determining users’ usage intentions of M-payments [16,18,39]. Zhou (2013) modified a trust-based adoption model and found that trust has significant direct and indirect impacts on the behavioral intention to use M-payment [20]. Moreover, trust has also been validated as an additional variable of UTAUT, which positively influences performance expectancy, consequently affecting user behavioral intentions to use M-payment [15]. Similar results have been found in other studies [35], including trust against perceived risk and uncertainty when adopting new technology [15,16]. Moreover, perceived risk combines uncertainty with the seriousness of the potential outcome [24], which negatively influences the multidimensional perceived benefits [14]. Thus, it can be summarized that trust has a positive impact on perceived benefits, which has also been supported by Khalilzadeh et al. (2017). Therefore, this study proposes the following hypotheses.

**Hypothesis** **7.**
*Trust has a positive effect on the behavioral intention to adopt M-payments during the COVID-19 pandemic.*


**Hypothesis** **8.**
*Trust has a positive effect on performance expectancy to adopt M-payments during the COVID-19 pandemic.*


**Hypothesis** **9.**
*Trust has a positive effect on perceived benefits to adopt M-payments during the COVID-19 pandemic.*


#### 3.2.5. Perceived Security (PS)

Perceived security is defined as “the degree to which a customer believes that using a particular M-payment procedure will be secure” [40]. In terms of conducting a financial transaction, lack of security—perception of security against the risk associated with mobile transactions—is one of the most frequent reasons of users refusing to adopt M-payments [41]. Previous studies have proved that perceived security is an important factor determining whether users will adopt M-payments [2,3,42]. Johnson et al. (2018) found that perceived security has the most significant positive impact on a user’s intention to adopt M-payment [43]. Moreover, perceived security significantly increases users’ trust by protecting users from transactional uncertainties and risks [15,44]. Shao et al. (2018) verified that security is the most significant antecedent of customers’ trust towards affecting usage of M-payment in both male and female groups [16]. Therefore, perception of perceived security of M-payment, considered as an extra variable of UTAUT, is a crucial guarantee for establishing users’ trust in using M-payment under a pandemic. Accordingly, this study proposes the following hypotheses.

**Hypothesis** **10.**
*Perceived security has a positive effect on the behavioral intention to adopt M-payments during the COVID-19 pandemic.*


**Hypothesis** **11.**
*Perceived security has a positive effect on trust to adopt M-payments during the COVID-19 pandemic.*


### 3.3. Research Model

Based on the above hypotheses, all measurement items were adapted from previous studies [4,8,11,14,15,16,19,24] and have been reasonably modified to correspond to the research purposes to explain the mental and technological factors affecting users’ behavioral intentions with regard to adopting M-payments under the COVID-19 pandemic. Specifically, users’ adoption intentions of M-payment under COVID-19 pandemic is conjointly determined by the variables from the revised UTAUT model (for explaining users’ technological perceptions) and perceived benefits, (as the variable of MAT, representing users’ mental cognitions and psychological acceptance of using M-payment under pandemic conditions). The questionnaire is presented in the Appendix A. Moreover, this study revises the UTAUT model, integrating performance expectancy, effort expectancy and social influence with additional variables, perceived security, trust and perceived benefits from MAT to establish a research model, depicted in Figure 1, with the proposed hypotheses relations.

## 4. Methodology

### 4.1. Measurement

In order to validate the proposed conceptual model and examine the research hypotheses, the online questionnaire survey was designed and applied to data collection. Specifically, the questionnaire consisted of two parts. The first part contained respondents’ demographic data with close-ended questions, consisting of gender, age, education, occupation and M-payment experience. The second part was developed by implementing constructs and items from previous hypotheses, consisting of 27 measurement items as indicators to explain perceived benefits, performance expectancy, effort expectancy, social influence, trust, perceived security and behavioral intention. In order to reduce confusion and save time for the participants [45,46], a five-point Likert scale (from 1 to 5, representing “strongly disagree” to “strongly agree”) was applied for representing the items of each construct.

The main survey target of this research was smartphone users who used or intend to use M-payment serviced in China during the COVID-19 pandemic. In order to avoid the impact of culture and language differences, the questionnaire was translated into the Chinese language by a professional translator, and then reversely translated into English, followed by confirmation of the translation equivalence. The questionnaire data were collected from a Chinese social media platform, named Wechat, for a three-week period during the height of the COVID-19 pandemic in China, from 11 March 2020 to 31 March 2020.

### 4.2. Data Demographic Characteristics

According to the N: q rule proposed by Jackson (2003), an ideal sample size-to-parameters ratio would be higher than 20:1 [47]; therefore, the sample size of this study should be higher than 140. This study dispatched a total 1000 online questionnaires via Wechat, 864 data were collected on 1 April. After removing the answers with missing values, a total of 739 valid questionnaires were accepted, achieving a final response rate of 73.9%. According to the guideline from Ryans (1974) [48], the Kolmogorov–Smirnov test was applied for verifying the nonresponse bias of the sample by comparing the groups between males and females. The demographic distribution of the sample was 45.74% male and 54.26% female; 53.86% of participants were in the age bracket between 21 and 30; 61.71% of participants held bachelor’s or college degrees (this group is more active on social media and so more likely to respond to the questionnaire) [49]; employees and students were the two main groups of participants, with percentages of 43.03% and 23.68%, respectively; 56.16% of total responses used M-payments at least one time per day and 93.78% at least one time per one week during the COVID-19 pandemic, which is in accordance with a report from Ipsos (2020) expressing that the penetration rate of M-payment among mobile Internet users in China (those who have used M-payment in the last three months) is 96.9% [50]. The reason of this high rate of adoption of M-payment during the pandemic can be summarized as follows. Firstly, based on the restrictions from the Chinese Government [10], due to daily transactions using contact being restricted during the COVID-19 pandemic, people tended to complete the transactions in a contactless way. Secondly, according to the suggestions and recommendations from to government and WHO [7], avoiding contacts among peoples is an efficient way to reduce the transmission risk of COVID-19. Thus, M-payment had been widely adopted by customers and retailers for general transactions. Thirdly, M-payment apps were applied to track users’ health statuses during the pandemic, such as Alipay Health Code being assigned a color code (green, yellow or red) to indicate users’ health statuses. Therefore, M-payments are dramatically adopted by smartphone users in China not only to support daily transactions, but also to confirm their health statuses during the COVID-19 pandemic. Specific sample demographics are listed in Table 2.

## 5. Data Analysis

The covariance-based structural equation modelling (CBSEM) technique was conducted for quantitative data analysis. SPSS 17 and AMOS 22 were applied in this study, through the two-step approach suggested by Anderson and Gerbing (1988), including validating the measurement model and testing structural model. The maximum likelihood estimation was conducted in the model assessment [51].

### 5.1. Measurement Model

A measurement model aims to assess fitness between indicators and latent variables. Exploratory factor analysis (EFA) was applied to examine the construct reliability, and a standard method factor analysis, confirmatory factor analysis (CFA), was applied to assess the convergent and discriminant validity of the measurement model. All seven hypothesized latent constructs in the CFA model were allowed to covary and were determined by related measurement items as reflective indicators.

Construct reliability was tested by Cronbach’s alpha. As presented in Table 3, all Cronbach’s alpha values of latent variables are in the range of 0.807 to 0.897, all exceeding the 0.70 suggested by Nunnally and Bernstein (1994) [52], which means that construct reliability has been demonstrated.

Convergent validity was assessed by standardized factor loading of all sample items. Table 3 shows that all items loadings are in the range of 0.807 to 0. All loadings are ideally greater than 0.70 [53], which demonstrates eligible convergent validity of the measurement model. Moreover, completed convergent validity was assessed by Composite Reliability (CR) and the average variance extracted (AVE) criteria. As shown in Table 4, the constructs have CRs in the range 0.811 to 0.898, all above 0.7 [54]. Meanwhile, all constructs have AVEs in a range of 0.589 to 0.688, which all meet the suggestions by Fornell and Larcker (1981) (that AVE should be higher than 0.5) [55], which means the latent variables explain more than half of the variance of the indicators. Therefore, the consistency of measurements among the indicators and latent variables has been proved.

Discriminant validity reflects whether two factors are statistically different. It is evaluated by using two criteria. Firstly, according to Fornell and Larcker (1981), the square root of the AVE should be greater than all correlations between each pair of constructs [55]. Table 4 shows that for each factor the square root of AVEs is larger than its correlation coefficients between all latent constructs, which proves that each construct shared more variance with its associated indicators than with any other construct [55]. Secondly, all AVEs are greater than the maximum shared squared variance (MSV) [56]. Thus, the scales satisfy the criterion of discriminant validity suggested.

Meanwhile, the following criteria (the ratio of chi-square to degrees of freedom (X^2^/df) < 3, comparative fit index (CFI) > 0.9, goodness of fit index (GFI) > 0.9, adjusted goodness of fit index (AGFI) > 0.9, normalized fit index (NFI) > 0.9, Tucker–Lewis index (TLI) > 0.9, and root mean square error of approximation (RMSEA) < 0.05) were applied to evaluate the fitness of the model. Table 5 shows that all the model-fit indices of the measurement model (X^2^/df = 1.832, CFI = 0.979, GFI = 0.948, AGFI = 0.935 NFI = 0.959, TLI = 0.979, RMSEA = 0.034), respectively, exceeded the common acceptance levels, which demonstrates a qualified fitness of the measurement model.

Further, this study examined the potential common method bias by Harman’s one-factor test (Podsakoff et al. (2003)) in SPSS [57]. The results show that the largest variance explained by an individual factor is 40.99% (<50%). The result confirms that none of the factors can individually explain the majority of the variance. Moreover, a CFA was applied to assess the fitness of a single-factor model (all items as the indicators of one factor) [58]. The results of the model fit present a poor fitness, which include v^2^/df = 15.999(>3), CFI = 0.603 (<0.9), GFI = 0.566 (<0.9), AGFI = 0.485 (<0.9), NFI = 0.588 (<0.9), TLI = 0.569 (<0.9) and RMSEA = 0.143 (>0.08). Therefore, both tests confirm that no common method bias appeared in this study.

The assessment results of the measurement model validate the construct reliability and convergent and discriminant validity of constructs satisfactorily. The constructs can be used to test the structural model.

### 5.2. Structural Model

The maximum likelihood estimation method and bootstrapping technique (500 samples and 95% significance level) were applied for assessing the structural model. Firstly, the model fit of the structural model was evaluated similarly to the measurement model. The results were presented in Table 5, which confirms a qualified goodness of fit of structural model.

Secondly, the variance (R^2^) of endogenous variables was assessed to evaluate the explanatory power of the structural model. As shown in Table 6 and Figure 2, the explained variances of performance expectancy (R^2^ = 0.48), perceived benefits (R^2^ = 0.28), trust (R^2^ = 0.44) and behavioral intention (R^2^ = 0.71) all exceed the recommended minimum value, 0.1 [59]. Thus, the structural model substantially explains the dependent variable.

Moreover, the results of hypotheses testing show that behavioral intention of adopting M-payments during the COVID-19 pandemic is the most significantly determined by performance expectancy (ß = 0.426; *p* < 0.001), followed by, perceived benefits (ß = 0.283; *p* < 0.001), social influence (ß = 0.277; *p* < 0.001), trust (ß = 0.234; *p* < 0.001) and perceived security (ß = 0.221; *p* < 0.001). Thus, Hypotheses H1, H2, H5, H7 and H10 are validated, respectively. However, the results show that effort expectancy (ß = −0.209; *p* < 0.001) has a negative effect on behavioral intention; therefore, H3 was rejected. Moreover, both effort expectancy (ß = 0.470; *p* < 0.001) and trust (ß = 0.401; *p* < 0.001) are statistically significant in terms of explaining the performance expectancy. Thus, Hypotheses H4 and H8 are confirmed. Meanwhile, the results illustrate that the perceived benefits are significantly influenced by social influence (ß = 0.272, *p* < 0.001) and trust (ß = 0.233, *p* < 0.001), respectively. Therefore, Hypotheses H6 and H9 are confirmed. In addition, H11 is also accepted by the result of perceived security (ß = 0.586; *p* < 0.001), significantly determining trust.

## 6. Discussion

Based on the data analysis results, ten of the eleven hypotheses were confirmed in this study, which demonstrates that the current study exhibits an appropriate adoption model to explain antecedents of users’ adoption intentions of M-payment under the pandemic.

Specifically, performance expectancy had the most significant positive impact on users’ adoption intentions of M-payments during the COVID-19 pandemic (Hypothesis 2), which corresponds to the vast majority of previous studies [37,60]. It can be confirmed that the utility and practicability of M-payment technology can improve users’ payment efficiency under emergency situations. Especially, M-payment provided a fast payment process without any direct or indirect contacts among people, significantly influencing users’ adoption intentions during the pandemic. Users will feel M-payment is a more useful and more reliable method than traditional payments to support their transactions under the pandemic.

Meanwhile, performance expectancy is significantly determined by effort expectancy (Hypothesis 4) and trust (Hypothesis 8), which is in accordance with findings from previous studies [2,35]. This study initially validated the effects of effort expectancy and trust on performance expectancy under the pandemic, which explains the absence of a confirmation of the simplicity and trustworthiness influencing the perceived functional utility when using M-payment under an emergency situation. Accordingly, the results support the accessibility and operability of the technology’s interface and function are positively formulate users’ performance expectancy; meanwhile, the reliability and trustworthiness of the technology’s services are essential to shape the high utilization of the technology under an emergency situation.

Moreover, the second largest significant effect on users’ behavioral intentions to adopt M-payments during the COVID-19 pandemic is caused by perceived benefit (Hypothesis 1). This result illustrates that perceived benefits correspond with individuals’ mental expectations related to contributions of M-payment under the pandemic. Specifically, perceived benefits, such as M-payment’s efficiency, not only influence users’ perceived technological perceptions, convenience and utility [14,24], but also increase perceived safety benefits by M-payment’s contactless characteristic. Concretely, users’ mental expectations are satisfied by perceiving more reliability and safety of using contactless payment to reduces contacts among people and maintains social distancing to decrease the COVID-19 transmission risk [7,31]. Thus, perceived benefits reflect users’ mental cognitions of technology’s features which can overcome a particular environmental issue, which in turn significantly influences users’ adoption intentions.

Meanwhile, under the condition of COVID-19 pandemic, perceived benefits as mental expectations are significantly influenced by social influence (Hypothesis 6) and trust (Hypothesis 9). The effects of social pressure and opinions of important, relevant people play an important role in influencing an individual’s mental expectations, affecting his/her behavioral intention [14]. When users receive recommendations from their close friends or families indicating that M-payment is beneficial for protecting their personal safety by avoiding contact with people during a transaction process to reduce the infection risk of COVID-19, they tend to consider M-payment as a helpful and valuable payment method. Moreover, trust was analyzed and found to have a significant effect on perceived benefits in this study. The reputation and trustworthiness of M-payments are potentially determined by the contactless advantage of M-payment in optimizing users’ experiences and supporting their safety during the COVID-19 pandemic, which emphasizes users’ perceived benefits towards adopting M-payments during the emergency situation.

Furthermore, social influence as the third important factor has a statistically significant impact on behavioral intention (Hypothesis 5), which means the opinions, recommendations and support from close relationships of users are essential in the formulation of users’ behavioral intentions to adopt M-payments during the COVID-19 pandemic. This result is supported by previous studies in normal situations [33,36]. Especially under the pandemic, people are relying more on the support and recommendations of important people in their lives—their family and close friends more easily influence their behaviors. Accordingly, the reputation of M-payment and word-of-mouth effect are considered crucial for attracting users’ adoption intentions of M-payment to formulate a new payment habit by the influence of the pandemic.

In addition, this study confirms Hypothesis 7 and Hypothesis 10—trust and perceived security have statistically significant effects on explaining users’ behavioral intentions of using M-payments during the COVID-19 pandemic. Specifically, consumers have developed trust in M-payment platforms through their reliable performance and mature legal framework protection, and so they worry less about financial risks to reap more benefits from the service [39]. Thereby, users’ adoption intentions are influenced by technological and privacy security and users’ trust from technological and mental perspectives [16,18,43]. Moreover, Hypothesis 11 also proved that perceived security significantly associates with trust. In this sense, perceptions of users’ perceived security could reduce uncertainty as well as crucially guaranteeing the M-payment performance to improve users’ trust of M-payment platforms [15]. It demonstrates that trust and perceived security have a significant association, and both factors conjointly determine users’ adoption intentions of M-payments under the pandemic. Furthermore, M-payments involve sensitive and personal data; therefore, it is necessary to ensure the reliability and credibility of M-payment platforms for securing transactions and protecting personal information [61]. Moreover, based on the security, trustworthiness and reliability of M-payment platforms, users can accept the records of their transaction times and locations during the pandemic to be utilized by governments and health institutions to track contacts among payment processes, for monitoring, updating and reporting the pandemic transmission status. Accordingly, users can clearly and opportunely be made aware of the virus infection situation among them, which positively influences their intentions to use M-payment during the COVID-19 pandemic to reduce the infection risk.

However, Hypothesis 3 was rejected in this study, which means easiness of understanding and handling M-payment systems does not have a direct impact on a user’s behavioral intentions to adopt M-payments during the COVID-19 pandemic. Similar results are supported by previous M-payment studies [3,62]. The main reason for this result is because users have become accustomed to smartphone functions and become more skillful through their previous utilization of various applications on smartphones [60]. Meanwhile, under the COVID-19 pandemic, user behavior is determined more by other perceptions related to personal safety, such as reliability, utility, security, trustworthiness and benefits, which can provide multidimensional supports for protecting transaction processes during a pandemic. Thus, the easiness of using M-payment is a less critical or surmountable factor determining users’ adoption intentions during the pandemic.

## 7. Theoretical and Practical Implications

### 7.1. Theoretical Implications

This study contributes three main theoretical implications. First, this study was empirical and examined the factors affecting users’ adoption intentions of M-payments under the pandemic situation, which is absent from evaluations of previous studies. Consequently, the study dramatically enriched the literature of technology adoption during a pandemic. Specifically, this study illustrates a worthwhile direction to understand users’ adoption intentions by not only examining users’ perceptions from technological perspectives, but also assessing users’ mental expectations. Moreover, users’ technological and mental perceptions of technology are significantly influenced by the emergency situations. Therefore, this study provides a future sight for relevant research to analyze new technology adoption from technological and mental perspectives conjointly and corresponding with the specific situation, especially for emergency situations.

Second, this study integrated the UTAUT model with perceived benefits from MAT and two extra variables, perceived security and trust, which significantly contributes to the theoretical development and framework coordination of the emerging literature on information technology adoption. Simultaneously, this study demonstrates a substantial contribution to the theoretical expansion of UTAUT and MAT by initially proposing and verifying new causal paths (PB → BI, SI → PB, TR → PB, TR → PE, TR → BI and PS → BI) and rejecting the path EE → BI for investigating the interactions of variables in the new comprehensive model. Therefore, the integrative research approach presented in this study can serve as a beneficial and valuable reference to modify and evaluate new adoption models for investigating novel technology adoption.

Third, this study initially focused on technology characteristics corresponding to the pandemic situation as a potential antecedent determining users’ mental and technological perceptions. Specifically, the contactless feature of M-payment avoids contacts during transaction processes and maintains social distancing, which improves the perceived multidimensional benefits of the users and optimizes their experience of using M-payments under the pandemic situation. Meanwhile, based on the disaster status of the COVID-19 pandemic, the effort expectancy became less important than other variables for determining whether users would adopt M-payment. Thus, it is important to consider whether a particular technology’s features can influence users’ interpretations of the perceived mental and technological benefits corresponding to particular situations or conditions to comprehensively explain technology adoption in an emergency situation.

### 7.2. Practical Implications

Moreover, four main practical implications are demonstrated in this study. First, the current research enhances the existing knowledge of adoption intention of M-payments in an emergency situation and enriches the understanding of how a pandemic changes users’ payment habits. It suggests that a pandemic might bring suffering to people or society. Furthermore, it can also facilitate the development of new technology that can bring benefits to individuals, organizations and society to survive in the emergency situation, which is valuable for relevant stakeholders to consider the pandemic to establish appropriate business strategies.

Second, this study could be valuable to start-up companies, policymakers, government bodies and private service providers who are interested in M-payment services. M-payment has become increasingly popular and provides useful services for efficient transaction processes, particularly in emergency conditions. In the context of a pandemic, M-payment can increase personal safety perception and maintain the stable development of business. Based on the finding from this study, as well as providing an easy-to-use operating application, relevant stakeholders should initially recognize the importance of M-payment in formulating users’ perceived benefits and design system attributes accordingly under the pandemic situation. Meanwhile, M-payment service providers should guarantee the compatibility, efficiency and security of transactions to meet customers’ requirements and match their lifestyles. In addition, enhancing the public impression of M-payment and stimulating a positive word-of-mouth social effect would improve the technology providers’ reputation in different situations.

Third, this study supports new technology providers with a comprehensive understanding of customers’ adoption intentions, determined by technological and psychological perceptions conjointly. Consequently, relevant stakeholders should focus on taking advantage of the features of technology (such as the contactless characteristic of M-payment) corresponding to its benefit to a particular situation (such as avoiding direct or indirect contacts to decrease COVID-19 transmission risk) in terms of maintaining service quality, reliability and efficiency to meet consumers’ physical and mental concerns and optimize their experience, thereby increasing acceptance among the target population.

Finally, the findings and results of this study could be applied as references for other online-to-offline (O2O) service industries in a pandemic situation. Relevant businesses could utilize the results to develop appropriate strategies that combine the benefits of technology characteristics with users’ technological perceptions and mental expectations to expand markets to adapt to different emergency situations and build better customer bases.

## 8. Limitations and Future Research

There are several limitations inherent in this study which need to be acknowledged. Firstly, the data collection was restricted to China during a particular period of the COVID-19 pandemic; the results may not be generalized to different countries and various situations. Future studies should replicate this model and collect data from different nationalities and consider specific benefits corresponding to particular situations. Furthermore, the research model can be examined through cross-cultural studies for better understanding of the variations in different cultural backgrounds.

Secondly, there were limited variables and interactions of the variables analyzed in this study—e.g., the variables selected in this study were mainly from a technology adoption aspect. Future research can put more effort into integrating the relations between variables, such as social influence affecting perceived security [15] and use technological indicators with the variables from a health and risk aspect. Meanwhile, in order to gain a deeper understanding of the mental and technological factors affecting adoption intentions with regard to novel technology, future research can incorporate research models with other variables, such as a cultural moderator, satisfaction, etc., which are also recommended in previous studies [42,63,64].

Thirdly, as the data collection period was limited, and that data were homogeneously distributed and collected through Wechat (a mobile social media application in China), in this study, the data collection process is recommended to chronically and integrally cover the users from different areas (urban and rural areas) over a different period of using M-payment in various patterns (online and offline surveys).

Finally, there was no distinction between the types of M-payment patterns (such as SMS, NFC and QR), platforms of M-payment (such as Apple pay, Samsung pay, Wechat pay, and Alipay) and patterns of electronic transaction (such as, electronic transaction via computer, electronic transaction via mobile device). Therefore, a future study can focus on distinguishing the different payment methods or payment platforms of M-payment techniques in accordance with specific research objectives.

## 9. Conclusions

In conclusion, we proposed a theoretical adoption model integrating UTAUT with perceived benefits from MAT and two additional variables, trust and perceived security, to appropriately explain the mental and technological factors affecting users’ behavioral intentions of adopting M-payment during the COVID-19 pandemic in China. This research model provided extensive explanatory power when explaining that users’ payment habits had changed due to the influence of pandemic, and that adoption intentions of M-payment were determined by technology perceptions and mental expectations conjointly. Performance expectancy, perceived benefits, social influence, trust and perceived security are significant in facilitating users’ adoption intentions of M-payments during the COVID-19 pandemic. Specifically, the contactless characteristic of the M-payment technique is beneficial in maintaining social distancing and protecting personal safety under a pandemic. This study also explored new causal relationships and found that perceived benefits are significantly determined by social influence and trust. Moreover, performance expectancy is influenced by effort expectancy and trust, towards explaining users’ behavioral intentions of using M-payments during the COVID-19 pandemic.

Furthermore, this study provides several significant theoretical and practical contributions on on investigating novel technology adoption in a particular situation, which contributes to the knowledge and understanding of the extension of the UTAUT application, explaining that users’ payment habits have changed because of the pandemic and adoption intention of M-payment is determined by users’ technological perceptions and mental expectations. In addition, this study recommends that researchers and relevant stakeholders focus on a particular characteristic of M-payments that corresponds with the pandemic, which can influence the perceived mental and technological benefits of the user. Understanding users’ behaviors is an efficient way to analyze new technology adoption and develop an appropriate strategy for optimizing users’ experiences.

## Figures and Tables

**Figure 1 ijerph-18-01016-f001:**
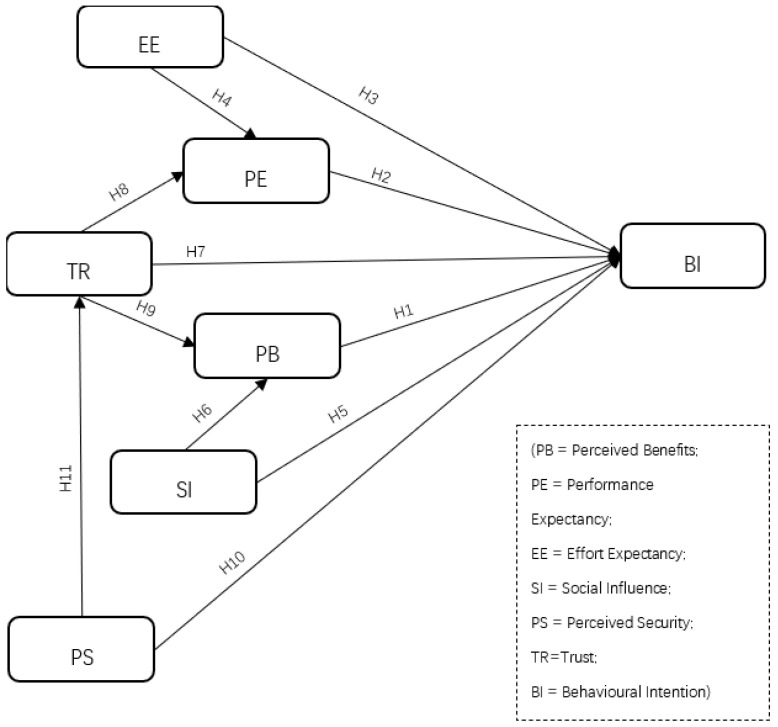
Research model with proposed hypotheses relations.

**Figure 2 ijerph-18-01016-f002:**
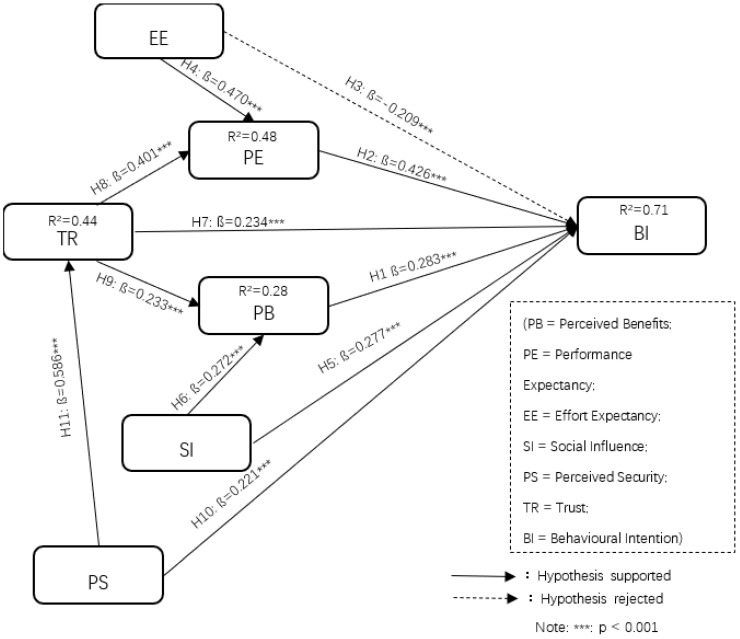
Hypotheses testing results.

**Table 1 ijerph-18-01016-t001:** Literature reviews related to Mobile payment (M-payment).

Studies	Theoretical Frameworks	Factors
**[15]**	UTAUT	Risk Security Trust Performance Expectancy (Hedonic and Utilitarian) Social Influence Effort Expectancy Self-Efficacy Facilitating Conditions
**[18]**	TAM	Perceived ease of use Perceived usefulness Trust Self-efficacy Subjective norms Personal innovativeness
**[14]**	Mental accounting theory	Technology anxiety Social influences. Multidimensional benefits (Convenient; Economic; Information security; Enjoyment; Experiential; Social) Attitudes towards using.
**[17]**	TAM	Perceived ease of use Perceived usefulness Subjective norms Attitude Perceived security
**[19]**	Expectancy-value theory, Task–technology fit	Characteristics Capability Task–technology fit Utilization Benefits

UTAUT: Unified Theory of Acceptance and Use of Technology; TAM: Mental Accounting Theory.

**Table 2 ijerph-18-01016-t002:** Demographic distribution of the sample.

Measures	Items	N	%
**Gender**	Male	338	45.74%
	Female	401	54.26%
**Age**	<21	170	23.00%
	21–30	398	53.86%
	31–40	80	10.83%
	41–50	29	3.92%
	>50	62	8.39%
**Education**	High school and lower	66	8.93%
	Bachler or collage	456	61.71%
	Master	194	26.25%
	PhD and above	18	2.44%
	Other	5	0.68%
**Occupation**	Student	175	23.68%
	Employee	318	43.03%
	Public Servant	47	6.36%
	Retiree	47	6.36%
	Unemployed	6	0.81%
	Freelancer	65	8.80%
	Other	81	10.96%
**Experience**	At least 1 time per 1day	415	56.16%
	At least 1 time per 1 week	278	37.62%
	At least 1 time per 2 weeks	37	5.01%
	At least 1 time per 1 month	7	0.95%
	Never use during the COVID-19 pandemic	2	0.27%

**Table 3 ijerph-18-01016-t003:** Item loadings and Cronbach’s alpha of structures.

Factors	Items	Loadings	Cronbach’s Alpha
**Performance Expectancy (PE)**	PE1	0.810	0.888
	PE2	0.850	
	PE3	0.792	
	PE4	0.812	
**Effort Expectancy (EE)**	EE1	0.813	0.897
	EE2	0.854	
	EE3	0.806	
	EE4	0.843	
**Social Influence (SI)**	SI1	0.805	0.894
	SI2	0.829	
	SI3	0.805	
	SI4	0.854	
**Perceived benefits (PBs)**	PB1	0.719	0.807
	PB2	0.828	
	PB3	0.751	
**Perceived Security (PS)**	PS1	0.773	0.848
	PS2	0.850	
	PS3	0.800	
**Trust (TR)**	TR1	0.771	0.878
	TR2	0.714	
	TR3	0.794	
	TR4	0.801	
	TR5	0.769	
**Behavioral Intention (BI)**	BI1	0.829	0.877
	BI2	0.799	
	BI3	0.777	
	BI4	0.797	

**Table 4 ijerph-18-01016-t004:** Descriptive statistics and correlation among constructs.

	CR	AVE	MSV	TR	PE	EE	SI	PB	PS	BI
**TR**	0.879	0.594	0.487	0.770						
**PE**	0.859	0.670	0.442	0.532	0.818					
**EE**	0.898	0.688	0.165	0.280	0.582	0.829				
**SI**	0.894	0.678	0.496	0.594	0.630	0.406	0.823			
**PB**	0.811	0.589	0.361	0.463	0.459	0.256	0.506	0.767		
**PS**	0.850	0.653	0.387	0.619	0.379	0.184	0.497	0.388	0.808	
**BI**	0.877	0.641	0.496	0.698	0.665	0.271	0.704	0.601	0.622	0.801

CR = Composite reliability; AVE = average variance extracted; MSV = maximum shared squared variance.

**Table 5 ijerph-18-01016-t005:** Model-fit indices of the measurement model and structural model.

	X^2^/DF	CFI	GFI	AGFI	NFI	TLI	RMSEA
**Recommended Value**	<3	>0.9	>0.9	>0.9	>0.9	>0.9	<0.08
**Measurement Model**	1.832	0.979	0.948	0.935	0.959	0.979	0.034
**Structural Model**	2.369	0.965	0.933	0.918	0.942	0.961	0.043

**Table 6 ijerph-18-01016-t006:** Hypotheses testing results.

Hypotheses	Relations	Estimate	T and *p*	Decisions
**Hypothesis 1: Perceived benefits have a positive effect on the behavioral intention to adopt M-payments during the COVID-19 pandemic.**	PB → BI	0.283	5.591 ***	Supported
**Hypothesis 2: Performance expectancy has a positive effect on the behavioral intention to adopt M-payments during the COVID-19 pandemic.**	PE → BI	0.426	8.059 ***	Supported
**Hypothesis 3: Effort expectancy has a positive effect on the behavioral intention to adopt M-payments during the COVID-19 pandemic.**	EE → BI	−0.209	−4.712 ***	Rejected
**Hypothesis 4: Effort expectancy has a positive effect on the performance expectancy to adopt M-payments during the COVID-19 pandemic.**	EE → PE	0.470	13.35 ***	Supported
**Hypothesis 5: Social influence has a positive effect on the behavioral intention to adopt M-payments during the COVID-19 pandemic.**	SI → BI	0.277	6.416 ***	Supported
**Hypothesis 6: Social influence has a positive effect on the perceived benefits to adopt M-payments during the COVID-19 pandemic.**	SI → PB	0.272	8.057 ***	Supported
**Hypothesis 7: Trust has a positive effect on the behavioral intention to adopt M-payments during the COVID-19 pandemic.**	TR → BI	0.234	4.254 ***	Supported
**Hypothesis 8: Trust has a positive effect on performance expectancy to adopt M-payments during the COVID-19 pandemic.**	TR → PE	0.401	11.242 ***	Supported
**Hypothesis 9: Trust has a positive effect on perceived benefits to adopt M-payments during the COVID-19 pandemic.**	TR → PB	0.233	6.306 ***	Supported
**Hypothesis 10: Perceived security has a positive effect on the behavioral intention to adopt M-payments during the COVID-19 pandemic.**	PS → BI	0.221	4.493 ***	Supported
**Hypothesis 11: Perceived security has a positive effect on trust to adopt M-payments during the COVID-19 pandemic.**	PS → TR	0.586	14.664 ***	Supported

PB = Perceived benefit; PE = performance expectancy; EE = effort expectancy; SI = social influence; PS = perceived security; TR = trust; BI = behavioral intention; ***: *p* < 0.001.

## Data Availability

The data that support the findings of this study are available from the corresponding author upon reasonable request.

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
