# Peer review of "How Does the Pandemic Facilitate Mobile Payment? An Investigation on Users’ Perspective under the COVID-19 Pandemic"

_ijerph, 2021, doi:10.3390/ijerph18031016_

Round 1
Reviewer 1 Report
Title: How does the pandemic facilitate mobile payment? An investigation on users’ perspective under the COVID-19 pandemic
The purpose of this paper is to comprehensively investigate the technological and mental factors affecting users' adoption intention of M-payment under the COVID-19 pandemic by integrated UTAUT and MAT perspective. It’s a good point to investigate how does pandemic facilitate mobile payment? However, this study should be revised for publication. Some suggestions are shown below:
Major:
- We know that China is the most popular countries using mobile payment. Because of the COVID-19 pandemic, the rate of Chinese people’s mobile payment has increased from 73.4% to 85.3%, so this research worthy to discuss. Moreover, the author adopts the unified theories of acceptance and use of technology (UTAUT) and mental accounting theory (MAT) to discuss the behavior of users adopting mobile payment, but having the samples like 56.16% (at least 1 time per 1 day) and 93.78% (at least 1 time per 1 week), which are all heavy users. It is inappropriate to explore the purpose of this research through such samples.
- This research framework did not list the factors that should be considered in the context of major disaster status such as the COVID-19 pandemic. For example, in the first quarter of 2020, countries have adopted lockdowns to prevent the spread of the virus. The ecommerce growth rate has also substantially increased the willingness and proportion of mobile payment, which cannot be explained by the MAT perspective. Therefore, the author's research framework cannot explain in the context of major disasters such as the COVID-19 pandemic.
Minor:
- Introduction: This study didn’t explain clearly why incorporated UTAUT and MAT model can comprehensively investigate mobile payment adoption behavior under the COVID-19 pandemic?
- Literature: This study explained clearly the concept of UTAUT and MAT, but it didn’t tell us why incorporated UTAUT and MAT model was suitable for explaining the research question.Otherwise, the study may tell us the restrictions or remedies in the COVID-19 pandemic in China to allow readers a clear understanding of the situation at the time to support this research to solve this type of problem.
- Development of hypotheses and research model: very clear.
- Methodology: see the Major 1.
- Data analysis: very clear.
- Discussion: This study didn’t explain why H3 is insignificant effect on user’s behavior intention to adopt mobile payment.
- Theoretical and practical implications: This study didn’t explain why H3 is insignificant effect on user’s behavior intention to adopt mobile payment.
Author Response
Dear reviewer,
We would like to thank you for the advantageous recommendations and comments. we have carefully reviewed each comment from your feedback, and did relevant editing and improvement of our paper based on the constructive comments and suggestions.The Point-to-Point response to your comments is uploaded as attachment. Please see the attachment.

Reviewer 2 Report
The increased use of virtual payments would be a highly expected result in the context of a dramatic increase in electronic platforms for carrying out transactions combined with a pandemic situation, where contacts have to be reduced to a minimum number. Therefore, this article becomes important due to the collection of empirical evidence of this expectancy as well as organizing theories that may contribute to understand users’ adoption intention of mobile payment in this particular pandemic situation.
The structure of the article is chosen correctly and it exhibits a logical concern. However, I have a few recommendations about the article:
I could not see if, instead of a mobile phone, it was used a computer to perform electronic transaction it would affect this study.
Regarding the sample, I was concerned with the lack of explaining the initial number of 1000 people to be included in the sample.
In addition, the authors do not indicate if they collected a balance sample because almost 62% had a Bachler or Collage and this may not be representative of Chinese users of mobile phone, table in line 324.
In line 505 the rejected hypothesis is H3 that is related to effort expectancy and not H2 as indicated in the text.
The quantitative data analysis and all tests conducted follow previous literature.
Author Response

(The authors gave the same response as above.)

Round 2
Reviewer 1 Report
This paper still does not clearly answer the major 1 question. The collected data can't explain the users' adoption of mobile payment in the wake of COVID-19 pandemic.
Author Response
Dear reviewer,
Thanks for your comments and suggestions. We have carefully revised our paper based on your comment from the second round of review. The detail answer can be found in the attachment.
If there was any further requirement, please feel free to let us know.
Thanks again for your recommendations and consideration.
Kind regards!
